# Robust Design Optimization and Emerging Technologies for Electrical Machines: Challenges and Open Problems

**Tamás Orosz** [1,*] , **Anton Rassõlkin** [2,3] , **Ants Kallaste** [2] , **Pedro Arsénio** [4,5] , **David Pánek** [1] , **Jan Kaska** [1] and **Pavel Karban** [1]

1   Department of Theory of Electrical Engineering, University of West Bohemia, Univerzitni 26,
    306 14 Pilsen, Czech Republic; panek50@kte.zcu.cz (D.P.); kaskaj@kte.zcu.cz (J.K.); karban@kte.zcu.cz (P.K.)
2   Department of Electrical Power Engineering and Mechatronics, Tallinn University of Technology,
    Tallinn 19086, Estonia; anton.rassolkin@taltech.ee (A.R.); Ants.Kallaste@taltech.ee (A.K.)
3   Faculty of Control Systems and Robotics, ITMO University, Saint Petersburg 197101, Russia
4   EDP Distribuicao, Direction of Market Platform, R. Camilo Castelo Branco 43-7th Floor,
    1050-044 Lisbon, Portugal; pedro.arsenio@edp.pt
5   Uninova-CTS, FCT Campus, 2829-516 Caparica, Portugal
*   Correspondence: tamas@kte.zcu.cz

**Abstract:** The bio-inspired algorithms are novel, modern, and efficient tools for the design of electrical machines. However, from the mathematical point of view, these problems belong to the most general branch of non-linear optimization problems, where these tools cannot guarantee that a global minimum is found. The numerical cost and the accuracy of these algorithms depend on the initialization of their internal parameters, which may themselves be the subject of parameter tuning according to the application. In practice, these optimization problems are even more challenging, because engineers are looking for robust designs, which are not sensitive to the tolerances and the manufacturing uncertainties. These criteria further increase these computationally expensive problems due to the additional evaluations of the goal function. The goal of this paper is to give an overview of the widely used optimization techniques in electrical machinery and to summarize the challenges and open problems in the applications of the robust design optimization and the prospects in the case of the newly emerging technologies.

**Keywords:** electrical machines, robust design optimization; digital twins; 3D printing, transformers

## 1. Introduction

Optimization is an essential part of research, both in science and in engineering. In many cases, the goal of the design is the outcome of an optimization problem [1–5], as is the case for electrical machinery, which belongs to the most general branch of mathematical optimization problems, both in the case of transformers and rotating machines [2–8]. Due to its complexity, the design optimization of electrical machines is usually split into different sub-tasks [6,8,9]. The two main sub-tasks are the design and the (preliminary) design optimization of the electrical machine. During the design optimization stage, the engineers' goal is to find the optimal key-performance parameters with a feasible scheme for a given application by investigating different topologies and multi-disciplinary analysis of the machine [1–5,9,10]. Then, the most minute details of the machine are worked out during the final design stage. Here, many sub-domain optimization tasks are performed to fine-tune the final design [5,11–13].

A wide range of numerical methods have been used to solve electrical machine optimization problems since the first appearance of computers in the industry [6], from simple heuristics to sophisticated mathematical optimization methods. For instance, transformer optimization was one of the first applications of geometric programming [9,14], but other deterministic methods, such as gradient based, trust-region, and interior-point based, were used to solve electrical machine related problems [15–17]. Here, the final solutions may depend on the initial starting points, and engineers need to simplify some functions to fit within the limitations of the applied mathematical programming method [14]. Besides, the computation of some of these functions is computationally expensive and cannot be calculated without numerical integration. Evolutionary algorithms provide a flexible and accurate framework to overcome these problems, and many genetic and evolutionary based electrical machine optimization methods have been published in the literature [5,8,18–22].

However, the selection of the right bio-inspired algorithm for a given application or the process of tuning its parameters is not straightforward. This task has its own challenges [23]. Due to the "no free lunch" theorem of mathematical optimization [24], none of the evolutionary optimization algorithms can outperform the others if they are averaged over all possible problems. However, metaheuristic based optimization solvers always have to be benchmarked from the point of view of the specific electrical machine optimization problem. As there is a most appropriate metaheuristic solver for a specific application, this is an important consequence of the "no free lunch" theorem. It is a challenging problem; there are a wide range of applications, and the optimized electrical machine models have had higher and higher complexity during the last few decades. Currently, most of the published electrical machine optimization methods are based on some FEM calculation methodology, to consider more details and achieve higher precision on the calculation of the performance parameters [8,9,25]. Many modern design methodologies optimize the electrical machine at the system level, together with its application; e.g., Silvas et al. in [26] considered a hybrid electric vehicle application for electrical machine design optimization, and S.A. Semidey et al. in [27] took wider applications into account and considered the load profiles of the drives as well. However, the solutions of these complex, system-level models are computationally expensive. The calculation time of one solution is usually longer than the time spent sorting the results [28]. This aspect should be considered during the selection of the optimization algorithm and the framework, because many of them were designed to perform these sorting and other optimization sub-tasks better, which is not relevant if we are working with computationally expensive tasks. However, the automatic parallelization and minimization of the number of function evaluations are important goals during these optimizations [29,30].

Another important question is: How can we handle the robustness during the optimization? This is due to the fact that when we are dealing with a real application, the tolerances in the manufactured dimensions or in the material characteristics are unavoidable [8,31–34], and designers are looking for a solution that is least sensitive to the perturbations [28]. However, an accurate sensitivity analysis needs the proper knowledge of the gradients. Therefore, the search region should be sampled appropriately, while the number of calculations (samples) should be minimized simultaneously. This is a complex numerical problem in the case of high-dimensional, unknown, computationally expensive functions, where under-sampling of the optimized region can easily lead to the wrong optima. Figure 1 shows a one-dimensional and a two-dimensional function, where the robust optimum differs from the global optimum of the task. Figure 1a illustrates the effect of under-sampling, when during the first optimization step, the unknown objective function ($f(x)$) is calculated at five randomly selected points in the parametric space. Then, the shape of the given expectation measure is different from the original objective function and does not provide information from the position of the different optimal points. By the application of the proper surrogate modeling techniques, the objective function can be approximated and calculated at more points during an expensive optimization task, which can significantly reduce the calculation time and provide accurate solutions [8,30,31]. However, inappropriately selected surrogate modeling techniques can easily lead to false optima.

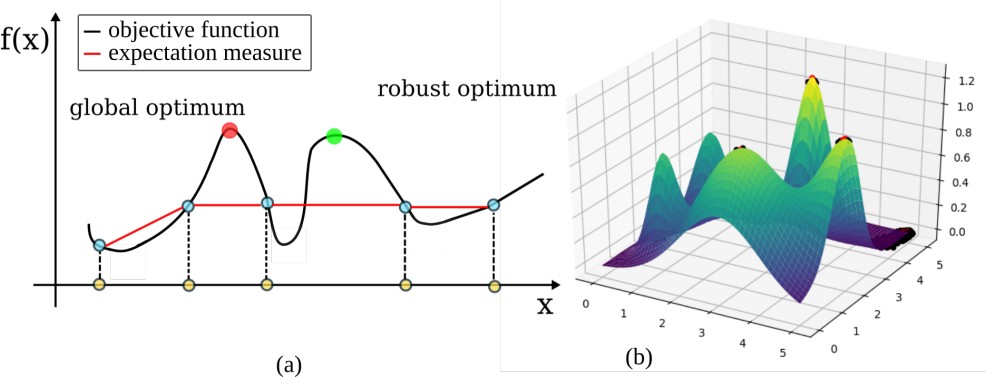

**Figure 1.** A one- and two-dimensional explanations of the robustness if the goal function(s) is (are) maximized. In (**a**), the yellow dots illustrate the problem of (under-)sampling during the gradient calculation (sensitivity analysis) of a numerically expensive optimization task. (**b**) shows the best solutions in the different tolerance classes in a robust optimization test problem (with Ārtap [35]).

This review aims to provide an overview of the existing optimization and numerical methods and tools, which can enhance the robust multiobjective optimization procedures, where the emphasis is placed on finding a robust frontier for electrical machine design related problems [36–38]. The second part of the paper highlights the role and possibilities of the robust design optimization based tools in the case of some recent technologies, 3D printed and superconductor based machines, and the role of robust design optimization in the development of electric vehicles and digital twin based modeling. This paper tries to summarize how the optimization problem changes if we want to find a robust optimum instead of searching for a global optimum, which does not exist in reality.

## 2. Robust Design Optimization Methods and Challenges

The design and optimization of electrical machines comprise a very wide topic in the literature, where many optimization methods and techniques are used to facilitate the optimization process and find better designs. However, finding a robust optimum instead of a global optimum is a different challenging mathematical problem, for which there is no unified mathematical definition of robustness in the literature.

The application of nature-inspired metaheuristics is very popular in electrical machine design and for different robust design optimization tasks. There are more than one hundred nature-inspired algorithms published in the literature [23,39–41]. However, all nature-inspired metaheuristics have algorithm-dependent parameters, which can significantly affect the performance of the algorithm. Moreover, it is still unclear and an open question in computer science what the best values or settings are and how to tune these parameters to achieve the best performance [23]. Moreover, the benchmarking and the selection of the appropriate nature-inspired metaheuristic is also a challenging open problem, due to the "no free lunch" theorem [24] of mathematical optimization. The extra difficulty here is that the calculation of a single individual or design is a computationally expensive task. Therefore, the application of modern numerical techniques, which can reduce the computational cost of one design's evaluation, should be considered during this challenging task.

### 2.1. Robustness in Machine Design and Manufacturing

Robustness is a general, desirable property in many engineering fields [42]. The design of an induction brazing process may well illustrate the need to consider various uncertainties (Figure 2). The optimized shape of the inductor has to be designed together with its control, considering the manufacturing tolerances to minimize the amount of waste products during the mass manufacturing process.

Due to its fast, clean, and contactless properties, induction brazing is a preferred method of many assembly processes [43,44]. The adjacent surfaces need to be heated, and the soldering material is melted on this surfaces. However, in the case of aluminum brazing, the difference between the temperature of the melting point of the soldering material and the aluminum is relatively small. To make a good joint, the temperature of the adjacent surfaces cannot differ by more than 10–20 K from optimal (Figure 2c,d), while the absolute value of the temperature reaches about 600 K from the initial room temperature within some tens of seconds. However, the temperature hot-spot is very sensitive to different kinds of positioning, manufacturing errors, the shape of the inductor, and the control of the excitation current [31,32,45]. To reduce the amount of waste produced during the mass manufacturing process, the tolerances and the control have to be designed and optimized together with the shape of the inductor.

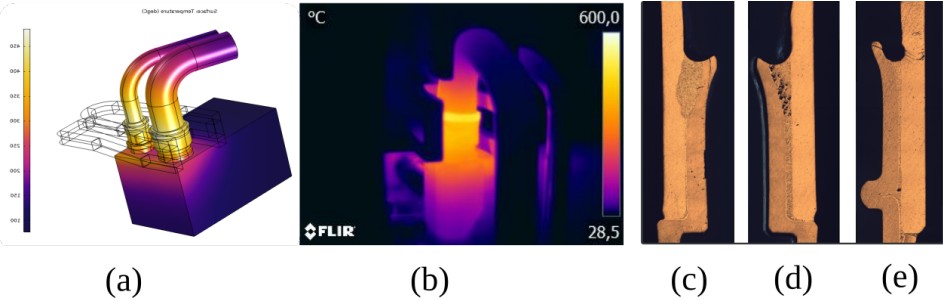

**Figure 2.** Coupled FEM model (**a**) and thermal camera image (**b**) of an induction brazing process design. The other three images show the possible errors during the brazing process: (**c**) dissolution, (**d**) porosity, and (**e**) gathering.

In the case of electrical machines, the design is robust if it is insensitive to small parameter changes and other uncertainties. These uncertainties can come from four different sources, and they are usually handled by a stochastic design approach [46,47]. Firstly, the most apparent and widely investigated source of these errors is the effect of the manufacturing tolerances on the electrical machine. Secondly—at the system or application level—the imbalances or non-linear properties of the electrical drives or changes in the operational or environmental conditions affect the performance of the machine [46,48–50]. For example, the proper handling of multi-harmonic excitations requires novel design methodologies. This issue concerns not only rotating machines, but also the insulation system design of an inverter fed by power transformer [51,52]. In some cases, it is necessary to consider the temperature of the environment to accurately calculate the demagnetization of the magnets [53]. Thirdly, the modeling or approximation errors can represent a different class of uncertainties. This kind of modeling error is unavoidable in the first, preliminary design stage, which deals with simplified machine models [9,54]. Another example is the parallel calculation of the time-dependent transient eddy current based quantities in electrical machines (multigrid-reduction-in-time algorithms [55]), like the parallel calculation of the steady-state parameter in the case of asynchronous [56] or permanent magnet machines [54] or the lightning impulse response calculation for large power transformers. In addition to the three already mentioned, Beyer [46] distinguished another source of uncertainty, the feasibility error. It considers what constraints of the design variables must be fulfilled. This kind of robustness is not independent of the first two cases. In the case of an electrical machine design, we also have limited knowledge about the exact material properties and boundary conditions. A wide literature on the numerical modeling of the B-H curves and demagnetization for electrical machine design exits [57–59].

### 2.2. Robustness Methods and Measures

Taguchi published the first and widely used methodology [60] that can systematically deal with the process uncertainties and tolerances. This methodology was originally designed for design processes, not for the design optimization of electrical machines. It uses the orthogonal design array

together with the method of the design of experiments [61] to select the optimal design [46,62–64]. This methodology is used for a wide range of applications, but it is possible to use the signal-to-noise ratio in a poor manner [62,65–68]. However, it is not easy to apply the traditional Taguchi method to design optimization problems [62]. This methodology was designed for discrete parameters, and it has difficulty handling wide and continuous parameters and a large number of constraints, which are essential during an electrical machine optimization process [46].

A wide range of papers deal with the sensitivity of a selected performance parameter of an electrical machine [69–71]. For instance, any mismatch between the motor and controller parameters will result in the deterioration of performance [8,69,72–74]. The thermal and cooling properties have a significant effect on their performance [75]. In the case of transformers, their leakage inductance and losses are non-linear functions of the excitation and geometry [7,9,14]. These examples are selected only to illustrate that a large number of sensitivity analyses should be considered during an electrical machine design. Generally, these sensitivity analyses consider the role of the uncertainties as a single variable problem. Various heuristics and artificial intelligence based methods were introduced to handle this kind of analysis, where the selected design variables are perturbed with the required tolerances to calculate the model sensitivities [36,76,77]. However, an electrical machine design problem generally means a non-linear optimization of many variables together, for which, if we improve one parameter, another one worsens. In the case of some simpler, transformer optimization methods, usually, more than 20 performance parameters are optimized [9].

The simplest methods directly use the sensitivity as the objective function of the optimization [62,78], as Belegundu [79] minimized this sensitivity information directly to obtain a robust solution, or [80], which used the second-order sensitivity to find robust solutions. However, these methods cannot be used for large-scale problems, such as complex electrical machine design tasks. Furthermore, the approximation of the objective function using the response surface methodology has been often used in robust optimization. Du and Chen proposed a similar meta-modeling technique: propagating model based uncertainty [81,82]. However, these methodologies have the risk of converging to a sensitive design because the sensitive part of the objective function can be ignored in the approximation process [62].

Generally, the design of an electrical machine is a multi-variable, multiobjective optimization task, where the designer has to select the best design from a set of contradictory objectives and parameters. In practice, it is hard to weight the importance of these different objectives. Therefore, a multiobjective formulation is one of the most accurate ways of handling these problems. The solution of the problem is not a single design, but a set of Pareto-optimal solutions. However, these objectives are not independent of a practical problem Figure 3. Here, none of the solutions is better than the others (Figure 4).

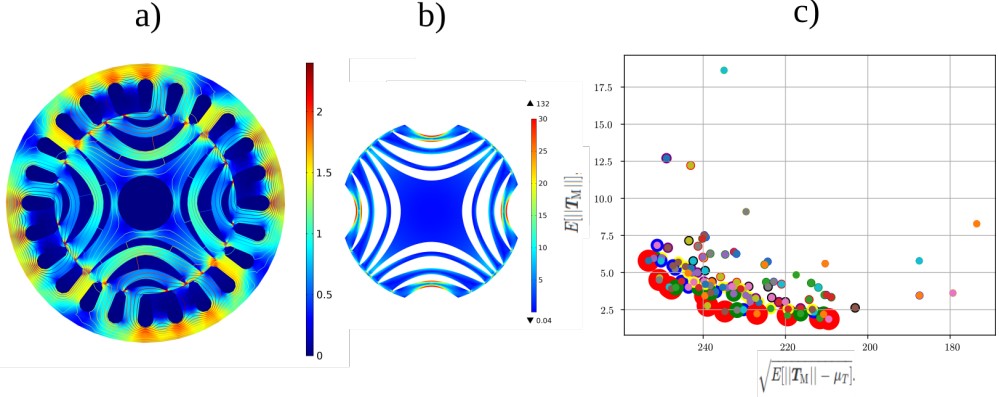

**Figure 3.** Optimization and multi-physical analysis of a reluctance machine, where (**a**) shows the flux density at the optimal shape of the machine, (**b**) depicts the von Mises stress, and (**c**) plots the distribution of the Pareto-optimal solutions. The y axes represent the average of the calculated torque, while the x axes represent the standard deviation of this torque from the average (torque ripple).

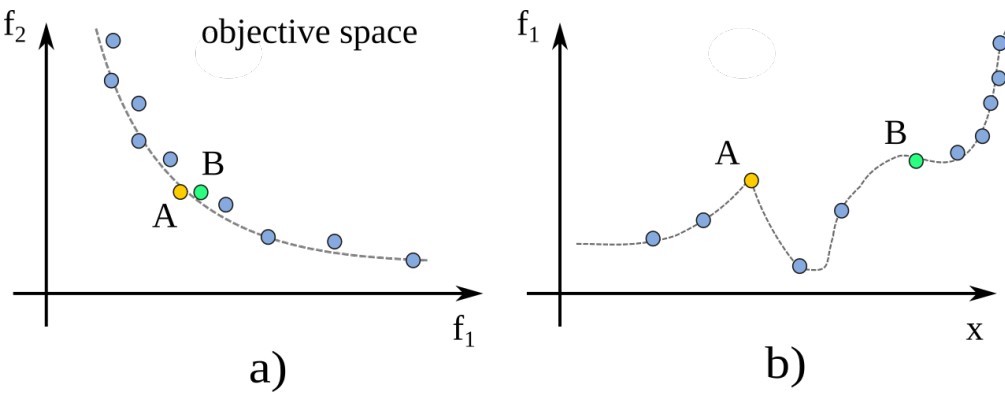

**Figure 4.** Illustration of a Pareto-front of a multiobjective optimization task. Two solution are highlighted by yellow and green color, where the yellow is sensitive and the green robust to the changes of the parameter x. In case (**a**) sensitive and robust solutions are close to each other and (**b**) solutions are distant from each other

Moreover, two neighboring points of the Pareto-plot can be very far from each other in the parametric space (Figure 4). The robust solution should be defined differently than in the case of a single optimization problem. Therefore, the Pareto-front of a robust optimization has four different positions compared to the original, multiobjective optimization task, as shown in Figure 5. Moreover, the computational complexity of a multiobjective optimization problem is much higher [36], because the tolerances should be checked for all of the examined objective functions.

The mathematical definition of robustness is not uniform in the literature [83]. There are several definitions, but they can be categorized as expectation or variance measures [38]. This terminology was introduced initially as the first- and second-type robustness by Deb [36].

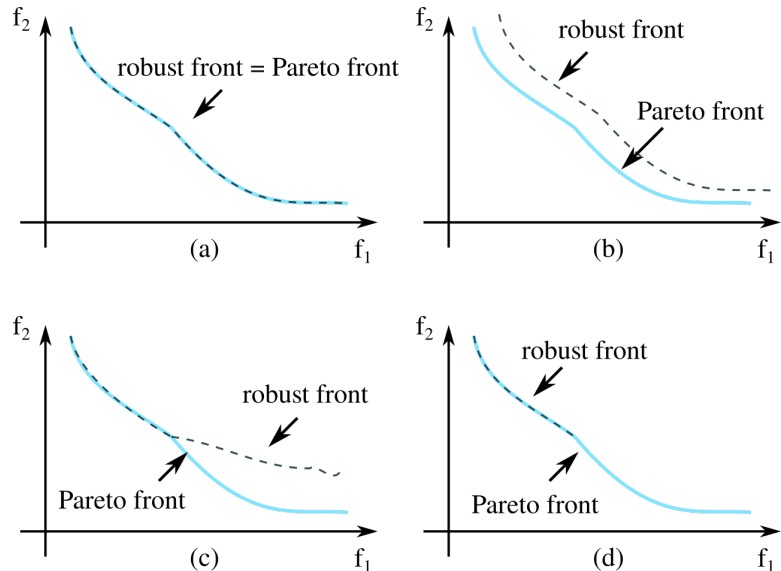

**Figure 5.** The four possible positions of the robust front compared to the Pareto-front of the multiobjective solution: (**a**) complete Pareto front is robust, (**b**) complete Pareto front is not robust, (**c**) a part of Pareto front is not robust and (**d**) a part of Pareto front is robust

The first kind of multiobjective robust design optimization formalism uses an expectation measure, e.g., an integration or averaging the values of the solution vector ($\vec{x}^*$) in the $\eta$ neighborhood of $f(\vec{x}^*)$:

$$Minimize \quad f^*(\vec{x}) = \frac{1}{|\beta_\epsilon(\vec{x})|} \int_{\vec{y} \in \beta_\epsilon(\vec{x})} f(\vec{y})d\vec{y} \tag{1}$$

$$x \in S \tag{2}$$

where $|\beta_{\epsilon(x)}|$ represents the hypervolume of the neighborhood, as illustrated in Figure 1. Several other forms of this expectation measure were introduced in the literature (e.g., explicit averaging or worst-case solution based estimation instead of the above-defined integral) to improve the performance of the applied optimization search methodology [38,83–87].

In this case, N times function evaluations should be performed around the neighborhood of the function. Due to the lack of analytical formulations for real-world electrical machine design problems, a Monte Carlo method based approach is usually used to randomly create N points around the selected solution. However, the electrical machine optimization is often numerically expensive, and this technique needs many extra function evaluations; moreover, there is a chance that the same value can be calculated more than once [36] due to the random behavior of these techniques. Advanced sampling methods (e.g., Latin-hypercube sampling) from the design of experiments [61,88] can help to overcome this problem.

Another problem with the calculation of this expectation measure is shown in Figure 1a. The continuous black line shows the values of the real objective function, and the red line shows the approximated values of the expectation measure from badly positioned and a small number of calculations. Stochastic gradient methodologies [89–91] work with the problem of how we can approximate the shape of an arbitrary, non-convex, non-linear function. This problem was firstly studied by Robbins and Monro [92]. Their fixed-point algorithm used implicit averaging and reduced step sizes to find local extrema of noisy function evaluations. The Polyak–Ruppert averaging made a key improvement on this technique by taking the averaged iterates along the optimization path into consideration [93,94]. This technique stabilizes and accelerates the convergence of these methods [95]. Many stochastic gradient methods use relatively large numbers of sampled data, which are sampled with or without replacement [96]. Re-sampling techniques can be disadvantageous in the case of expensive optimization problems, like electrical machine design.

In the second type of multiobjective robust design optimization (Figure 6), the variance measure based formalism can be defined in the following way:

$$Minimize \quad f(x) \tag{3}$$

$$subject \quad to \quad \frac{||f_p(x) - f(x)||}{f(x)} \leq \eta \tag{4}$$

$$x \in S \tag{5}$$

This methodology uses constraint functions to exclude the solutions, which are more sensitive than a previously defined $\eta$ threshold (for $\eta = 1$, we get the non-robust case).

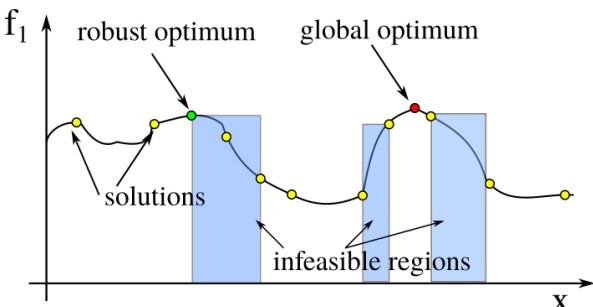

**Figure 6.** The infeasible regions in the case of a type II robustness. These regions are calculated from the (badly) sampled points of the objective function (yellow).

*2.3. Multi-Objective Optimization Techniques*

As previously shown, electrical machine optimization during a sensitivity analysis task is often considered as a single objective optimization problem. However, there are many conflicting goals in the background. These kinds of optimization methods can be handled better as constrained, single-objective problems. Several examples can be found for this kind of model and optimization techniques in the literature. Perhaps one of the most general applications in electrical machine design is solving the inverse design problems with these kinds of methods, like the preliminary design of transformers or rotating designs [9,14,97–102]. The advantage of this constrained single-objective formalism is that the mathematical programming based approaches can be used for the solution of these problems, like quadratic programming [103] or geometric programming [104]. Their formalism and the built-in modern interior-point method based convex optimization solvers can quickly find the solution and prove that the found solution is the global optimum or proof that its exist [15,105]. Moreover, these methods do not require an initial guess of the values or a reference solution to start the optimization. However, applying these methods requires an analytical model of the electrical machine, which fulfills the special requirements of the different mathematical programming tasks. This is not possible in any case, e.g., the case of a complex, finite element method based calculation.

However, in real-world design tasks, the objective functions are usually not only conflicting, they are incommensurable [41,106,107]. Therefore, in these cases, there is no unique solution that can minimize all of the given objective functions, so the solution of the task is a set of solutions, which follows the concept of Pareto-optimality [107,108]: a solution $x^*$ is Pareto-optimal if there is no other existing solution ($y$) such that $f_i(y) \leq f_i(x)$ for all $i = 1, .., m$ parameters of the solution and $f_i(y) < f_i(x)$ for at least one parameter of the solution, if the optimization task is to search for the minimal value of the feasible solutions in the decision space.

There are many metaheuristic approaches used for electrical machine design optimization [5,8,109–116], and these methods can be categorized in many different ways. Due to their different properties, these methods can be assigned more than one category [117–119]. One of the easiest ways is to categorize them as stochastic or deterministic approaches. Those methods that do not contain any randomly changed parameters are called deterministic approaches [8]. One possible categorization of some selected techniques is depicted in Figure 7. The different stochastic approaches are very widely used in electrical machine design, because these methods can be used for gradient-free search of the solutions, and they do not need an initial solution to converge to the global optimum [120]. These kinds of population based optimization methods, like Genetic Algorithm (GA) [121–123], Particle Swarm Optimization (PSO) [124], Differential Evolution (DE) [125,126], Covariance Matrix Adaptation Evolution Strategy (CMA-ES) [127], Firefly Algorithm (FA) [128,129], Cuckoo Search (CS) [130], etc., are successfully used for electrical machine design optimization [131].

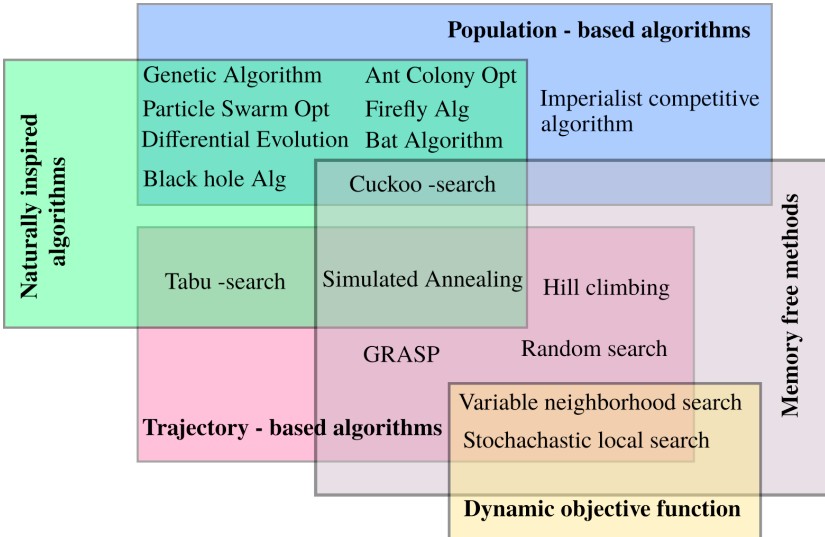

**Figure 7.** Some selected metaheuristic methods and their categorization.

The main difference between these algorithms is that they use different operators and methodologies to share the information between the selected parents and offspring. From this point of view, the single objective algorithms can be categorized in the following way [23]: Gradient-Guided Moves (GGM), Random Permutation (RP), Direction Based Perturbations (DBPs), Isotropic Random Walks (IRWs), and Long-Tailed, scale-free Random Walks (LTRWs). For instance, the PSO algorithm, which uses the DBP technique, needs the best individual's information to generate the new position of the particle. Meanwhile, the genetic algorithm does not care about which individual in the examined population has the best solution. This problem of selecting the best individual can lead to a complex question if we want to use PSO—or similar metaheuristics—for multiobjective optimization, where it is not possible to define a single best solution, only a leader set of the Pareto-optimal solutions. This problem can highlight why there exist many different multiobjective optimization variants (e.g. Adaptive Multi-Objective PSO (AMOPSO) [132], Multi-Objective PSO (MOPSO) [133], Speed-constrained Multi-objective Particle Swarm Optimization (SMPSO) [134], Optimized Multi-Objective PSO (OMOPSO) [135], Vector Evaluated PSO (VEPSO) [136], etc.) of the PSO algorithm in the literature (Table 1). These algorithms do not only use different approaches to handle the leader set with Pareto-dominance or $\epsilon$-dominance based archives [137] or a method that randomly selects an arbitrary element for the current best element, like in the multiobjective firefly algorithm [39]; however, the PSO algorithm—as many other single objective metaheuristics—has greatly developed from its first publication. The first and still widely used modification is the introduction of the constriction factor by Eberhart and Shi [40]. However, there are many modifications and improvements that have been introduced in the last two decades, from the time-varying inertia weight and velocity constraints and the gradient based [138] speed components to the multi-level, hierarchical PSOs [139]. In the field of the electrical machine design optimization, PSO represents an umbrella term, which hides the information about the PSO variant used. This information is usually not highlighted in the research nor review papers [3,5,8,19,51,140].

**Table 1.** Widely-used nature-inspired multiobjective optimization techniques.

| Technique | Name | Description |
|---|---|---|
| NSGA-II | Non-dominated Sorting Genetic Algorithm II | It is a widely used optimization method. It uses the elitist strategy with the crowding distance operator to preserve diversity and the efficient non-dominated sorting operator to select the Pareto-dominant solutions [141]. |
| NSGA-III | Reference point based Non-dominated Sorting Genetic Algorithm | This algorithm is designed for many-objective problems (more than two). It uses similar operators, like NSGA-II with reference points and the niche preservation operator, where the reference point can be associated with some solutions, and it keeps the solutions that are close to the reference point [142]. |
| $\epsilon$-MOEA | $\epsilon$-dominance based Multiobjective Evolutionary Algorithm | It uses the epsilon-dominance concept with an epsilon archiving strategy for limiting and sorting the Pareto-dominant solutions, which can be faster than the non-dominated sorting, and it can be advantageous in many cases. However, the sorting time is not relevant in most cases of numerically expensive electrical machine design problems [143]. |
| MOEA/D | Multiobjective Evolutionary Algorithm with Decomposition | This algorithm explicitly decomposes the problem into scalar optimization subproblems. It solves these subproblems simultaneously. At each generation, the population is composed of the best solution found so far for each subproblem. This algorithm can significantly reduce the computational complexity compared to NSGA-II [144]. |
| GDE3 | Generalized Differential Evolution | GDE3 uses the weak-dominance concept to select the Pareto-dominant solutions and an improved crowding distance operator and non-dominated sorting for the results [145]. |
| PAES | Pareto Archived Evolution Strategy | It uses a non-dominated bounded archive to maintain the Pareto-optimal solutions. There are three main versions: (1+1), (1+$\lambda$), and ($\mu$+$\lambda$)-PAES [146], where (1+1)-ES means that during each iteration, one mutant is created from one parent, and their union is used in the selection. |
| PESA2 | Pareto Envelope based Selection Algorithm | It uses a region based selection operator, instead of individual based ones, like NSGA-II. It requires only $\mathcal{O}(k \cdot n)$ comparisons for every hyperbox [147]. |
| SPEA2 | Strength based Evolutionary Algorithm | SPEA-2 uses the strength based diversity operator, the calculation and sorting time of which are more expensive than the case of NSGA-II; however, the diversity and convergence of the results can be better, which can be advantageous in the case of expensive optimization problems [148]. |
| IBEA | Indicator Based Evolutionary Algorithm | It uses a flexible integration of preference information. Therefore, an arbitrary performance indicator can be used for the search. It does not need any diversity preservation techniques; moreover, the population size can be arbitrary [149]. |
| MO -CMA-ES | Covariance Matrix Adaption Evolution Strategy | The adaptive grid archiving strategy, which was presented in PAES, is merged with the covariance matrix adaption evolutionary strategy. In contrast to most other evolutionary algorithms, the CMA-ES is quasi-parameter-free [150]. |
| OMOPSO | Optimized Multiobjective Particle Swarm Optimization | Uses crowding distance to filter out leader solutions and the combination of two mutation operators to accelerate the convergence of the swarm and $\epsilon$-dominance based archiving strategy [135]. |
| SMPSO | Speed-constrained Multiobjective Particle Swarm Optimization | It is very similar to the OMOPSO algorithm, and there are only three differences: the usage of the speed constriction factor and the polynomial mutation operator and velocity handling at the borders of the search space [134]. |
| NSPSO | Non-dominated Sorting based multiobjective Particle Swarm Optimization | It uses the main mechanisms of NSGA-II (crowding distance, non-dominated sorting), and the global leader is selected randomly from the leaders' archive. OMOPSO and SMPSO clearly outperform this variant [134]. |
| AMOPSO | Another Multiobjective Particle Swarm Optimization | It selects leaders from a non-dominated external archive. Three different selection are techniques used: Roundsby to preserve diversity, calledRandom to promote convergence, and called Prob, a weighted probability method [132]. OMOPSO and SMPSO outperform this method. |
| MOFA | Multiobjective Firefly Algorithm | Uses random weights to select the best from the Pareto-optimal solutions. Very quickly converges to the solution; however, it contains usually more than one function evaluation in every iteration step because the selected firefly makes one step with a new evaluation of a dominating one. This is a disadvantage in the case of expensive optimization problems [39]. |

The Multiobjective Evolutionary Algorithms (MOEA) can generally be divided into three main paradigms: the Pareto-dominance based MOEAs (like NSGA- II [141]), Indicator Based Evolutionary Algorithms (IBEAs, [149]), and decomposition based MOEAs. The Pareto-dominance based algorithms work on the principle of non-dominated sorting where each solution is assigned a rank based on its Pareto-dominance. Usually, the best solutions are given preference during the mating and/or the selection process (elitist strategy) to increase the convergence. Moreover, these methods employ strategies such as crowding distance (NSGA-II) [141] and weighted distance [151] to increase diversity. In contrast to these algorithms, the indicator based strategies (IBEAs) directly contain performance indicators to select the most appropriate offspring [149,152]. These performance indicators, like the hypervolume [152] and inverted generational distance [153,154], were originally designed to

measure the distance, the diversity of the found Pareto-front from the identical one. The third group contains the decomposition based algorithms, where decomposition is a procedure that breaks down the given problem into smaller pieces and then optimizes them sequentially or in parallel [107]. This paradigm is incorporated with many metaheuristics, like Tabu-search [155], simulated annealing [156], PSO, etc. [107].

Because electrical machine design is an expensive optimization task, during the algorithm selection, we should prefer those algorithms that use fewer function evaluations. Instead, they require more calculation time for sorting. For example, the Multiobjective FA (MOFA) [39] contains more than one function evaluation for the intensity between two individuals. If the intensity is more preferable, the actual firefly moves to a new position during the iteration step. This means 100 iterations in 100 individuals need more than ten thousand function evaluations, as opposed to an optimization with the NSGA-II algorithm, which makes only 10,000 function evaluations with the same setup.

The NSGA-II algorithm [141] is one of the most popular population based multiobjective algorithms in the industry [157], and the source code was published by Deb [141]. There are many minor implementation differences between the different, high-quality codes because there are many novel implementations, tricks, and minor improvements that have been published in the last decade [158–160]. Moreover, these tools (pymoo [161], DEAP [162], Inspyred [163], Platypus [164], Ārtap [29], PyGMO [165], jMetal/jMetalPy [166]) have different goals; some of them have been developed as a universal optimizer or a framework for rapid prototyping of new ideas in evolutionary computing, while some of them have been designed for high level parallelization and expensive calculations, where the calculation cost of a single individual is much higher than the consideration of the calculation speed during the sorting of the elements.

## 2.4. Measures for Evolutionary Algorithms and Uncertainties

The selection of the best or the most appropriate metaheuristic is a general need for electrical machine designers [5,8,167]. However, due to the no free lunch theorem of mathematical optimization [24,168], there is not a metaheuristic method that is superior to any other for all of the possible problems. One necessary consequence is that the different algorithms should be benchmarked and compared on a task that is similar to the given design optimization task [23]. This problem is still not widely investigated in electrical machinery papers. There are only a few open benchmark problems, like the multiobjective TEAM (Testing Electromagnetic Analysis Methods) benchmark problems [169–171], which provide a theoretical test problem for a multiobjective and robust homogeneous magnetic field design. Moreover, benchmarking and making high-level performance measures for evolutionary and genetic algorithms are still open questions [23].

There are many mathematical benchmarks that have been produced to test and compare the different optimization algorithms. These benchmarks usually contain and mimic the following difficulties that the optimizer has to handle [38,172]: slow convergence, a large number of local optima, a large number of variables, the dependency of variables, constraints, deceptive search spaces, flat search spaces, and uncertainties. There are different test sets to benchmark and compare the different optimization algorithms [173]. One of the most widely used is the Congress on Evolutionary Computation (CEC) competition benchmark sets [174]. Due to the lack of specific benchmark problems, these and similar general performance measures are usually used in the electrical machinery literature to select and compare the metaheuristics for a specific task, instead of using a similar, specific benchmark problem, as recommended by the no free lunch theorem of mathematical optimization [174,175].

Due to the difficulties of handling uncertainties, there are some robust design optimization test functions published in the literature [38,176–180]. Reference [38] proved the ineffectiveness of the normal PSO and genetic algorithms on these test problems. Replacing their objective functions with expectation measures, these problems can be solved with the application of the robust versions of the PSO and genetic algorithms. The solution of a robust alpine function is illustrated in Figure 1b.

There are many techniques that have been introduced in the literature that can increase the performance of the evolutionary algorithm based calculations without increasing the number of samples [37,181]. Parmee proposed an evolutionary algorithm based space decomposition method [182,183], which divides the search region into high- and low-performance regions as a function of the sensitivity of the examined parameter. This technique performs further optimization only in the high-performance regions. This technique can significantly decrease the computational cost. However, as shown by Wiesmann [86], nothing guarantees that the optimal solution is in the high-performance region.

Another possibility is explicit averaging over time, where explicit averaging means a form of resampling. Increasing the sample size is equivalent to reducing the variance of the estimated fitness function. Aizawa and Wah were the first to propose adapting the sample size during the run. They proposed to start the calculation with a relatively small sample size and increase the number of individuals with the number of generations [184,185]. Other adaptation methodologies can use a higher sample size for those individuals that have a higher estimated variance or simply calculate the fitness by averaging over the neighborhood of the evaluated individuals [48]. Another possibility to reduce the noise is implicit averaging because the area of a possible solution is sampled repeatedly, and this information can be used for an implicit averaging. This methodology does not need to recalculate these results. Moreover, it was proven mathematically that if the population size is infinite, the proportional selection is not affected by noise [48,49]. It is always an interesting question whether explicit or implicit averaging can reduce the computational complexity better. Genetic Algorithms (GAs) with a finite population size have been studied in many of the models created, which can optimize the sample size and the population size at the same time [186,187].

Many researchers suggested modifying the selection operator and using deterministic selection schemes in genetic and evolutionary algorithms to better handle the different types of uncertainties [48,49,188]. Markon et al. [189] suggested a threshold during the selection process in an evolutionary strategy. This methodology accepts the newly generated offspring only if its fitness is significantly better than its parent. Branke and Schmidt proposed a derandomization for better handling of uncertainties in the selection process [190,191]. Gutjahr et al. [192] proved that simulated annealing does not converge under a specific class of noisy environments. A modified, deterministic selection operator for simulated annealing can improve the performance under noisy optimization tasks [48].

Rakshit and Conar suggested a bee colony optimization algorithm, which uses the operator mentioned above and the selection and implicit averaging techniques together [188]. In their other paper, they introduced four principles for selecting an evolutionary algorithm in a noisy environment [193]: Firstly, the sample size during the optimization should be adapted in time, increased exponentially with the generation number. Secondly, they proposed to use the above-mentioned modified, deterministic selection schemes [194,195]. Thirdly, they proposed to use a clustering approach. Finally, they developed a robust crowding distance scheme that can work better in noisy environments [193].

*2.5. Methods for Computational Cost Reduction*

Electrical machines' accurate modeling requires solving partial differential equations in many physical domains (electrical, mechanical, thermal, etc.) simultaneously. Moreover, the number of possible designs is very large. If we are thinking back to the preliminary design problem of a large power transformer, where 20 design parameters were optimized simultaneously, there exists millions of possible designs for the same specification [9]. The design task can be more expensive numerically in a system-level design of a rotating machine, where the electrical machine and the control are designed together. To reduce the computational demand of these tasks, a wide variety of meta-modeling techniques can be employed to evaluate the fitness function and avoid complex numerical simulations [48,196,197]: response surface methodologies [198,199], multilayer

perceptron [194,200–203], radial basis function neural networks [204,205], kriging models [206,207], Gaussian processes [208], support vector machines [209–211], fuzzy logic [212].

However, it is difficult to make a meta-model from an unknown expensive optimization problem, which approximates the function of the whole region (Figure 8). Generally, the kriging or Gaussian process approximations can evaluate the function with sufficient accuracy in a given region of the function. However, they cannot approximate the minimum functions or make false global minimums, which is not part of the original function. To overcome these problems, these approximate functions are usually used together with the original function, and if the error indicator is small enough, the calculation uses the approximate value [171,202]. Kriging based techniques are widely used for electrical machine design problems [109,213–216]. For example, Bittner et al. [217] used it together with PSO to optimize a permanent magnet synchronous machine for hybrid and electric cars, and Woo [218] used it to find the optimal shape of the rotor structure. In [219], the authors explored the importance of achieving an appropriate balance between exploration and exploitation. They proposed two kriging based strategies to approximate the fitness functions of expensive electromagnetic design problems with high accuracy and low computational effort. Many electrical machine design paper use the response surface methodologies [8,21]; however, as previously mentioned, they can converge to a sensitive solution, because the sensitive part of the objective function can be ignored [62].

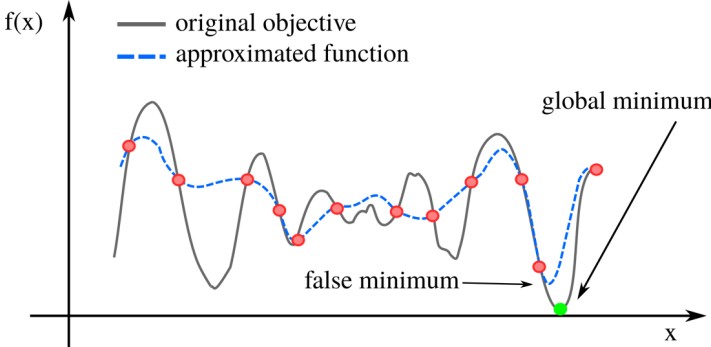

**Figure 8.** An example of a false minimum in the surrogate model.

## 3. Numerical Modeling and Novel Technologies and Materials

### 3.1. Drives and Electric Vehicles

The electrical machine is one of the main component in many different electrical energy drive and conversion systems. These systems use more than 53% of the consumed electricity in the different sectors of the industry [220], which means about 6040 Mt of $CO_2$ emissions per year. Due to global sustainability, the electrical drive systems have to fulfill more and more rigorous requirements, not only physically and technologically, but also environmentally [167].

Each electrical drive system, besides electrical machines, also includes power electronic devices, control systems, and other protection and mechanical subsystems, as illustrated in Figure 9. The integral optimization of electrical drive systems to productively maximize overall efficiency typically includes the application of more efficient and well-sized components [221]. Both the research and industry communities are highly interested in the design and optimization of electrical drive systems, especially in the field of Electric Vehicles (EVs). Research practice shows that the system-level design part plays a significant role in development. Due to the non-linearity of the whole system, it is more valuable to follow the optimal performance of the whole drive system rather than pick up excellent separate components (e.g. power electronics, electric motor, or gear), and usually, assembling of such individually optimized single components into a complete drive system cannot ensure the best performance [222]. For example, extra harmonic losses can appear in the electrical machine due to the harmonic distortion in power electronic devices. The loss analysis methodology must be applied to the

holistic drive system to ensure that reducing losses in a single component will not have a negative effect on the efficiency of the other parts. and vice versa [223]. Due to the high time requirements and costs, only a limited number of electrical drive system components have been redeveloped recently [224]. Rather, the actual components are slightly modified and adapted to new applications. G. Lei et al. [222] suggested a five step design optimization framework for the EV electrical drive, as follows:

Step 1: determination of electrical system requirements (e.g. cost, efficiency, speed range, controllability, power density, electrical machine torque, etc.);

Step 2: topology selection;

Step 3: design of the electrical machine and/or power electronics device;

Step 4: optimization of designed parameters.

Step 5: evaluation of the whole electrical drive system's performance, including the steady-state performance and dynamic response.

However, this methodology decomposes the optimization task into a separate machine and electronic device design, as two parallel, independent optimizations. Therefore, this methodology can give a good practical design in a reasonable time, but cannot ensure (mathematically) that the found optimum is the global optimum of the whole electrical drive system.

In other studies by G. Lei et al. [167,225–227], a robust approach for the system-level design optimization of the electrical machine and drive system based on Design For Six Sigma (DFSS) was presented. DFSS allows using empirical data, and it is based on the use of statistical tools. Certain case studies (including multiobjective and system-level optimization) on a PM transverse flux machine with a soft magnetic composite material core were investigated by the authors. Based on the studies, it may be highlighted that the electrical drive system's reliability may be consequently improved by a robust design optimization approach. This fact may extend the industrial application area of the drive system and benefit manufacturing. However, for this DFSS methodology, as for the Taguchi method, designed for discrete parameters, it is difficult to handle wide and continuous parameters and a large number of constraints. S. Kalt et al. in [224] presented an application based design method that focuses on implementation of the electrical machine design process for EV electric drive system requirements. As stated in the research, such an application based design methodology allows shifting the highest electrical machine efficiency regions to the real operating points of the target EV. As a result, a customized electrical machine design for a specific EV is presented.

Usually, design optimization is quite often based on the application of an electric drive system; for example, Reference [228,229] considered the driving cycles of an EV as a reference. The paper by M. Degano et al. [228] dealt with a high-speed PM assisted synchronous reluctance motor design and its optimization according to U.S. standard driving cycles (city: Urban Dynamometer Driving Schedule (UDDS); highway: Highway Fuel Economy Driving Schedule (HWFET)) to evaluate the most representative operating points of the studied application. The main results of the research were the most efficient operation areas and electromechanical specifications of an electrical machine. In order to define optimization inputs, a geometrical parameter analysis of the permanent magnet synchronous machine was performed. L. D'Angelo et al. [229] considered geometrical defects that are caused by electrical machine manufacturing errors, ambient conditions (including traffic and weather), and possible differences in driving styles to obtain robustification. The object of the study was a permanent magnet synchronous machine, where the rotor's and permanent magnets' geometries varied. Three different scenarios were observed, and the optimal electrical machine configuration for each scenario was suggested; however, the optimization results were very similar to the initial configuration, as the nominally optimized configuration was never robust.

Optimization may also be implemented in the case of autonomous EVs, such as [230], e.g., for distribution path optimization. In [231], to avoid the confidence that uncertain transport time data are assumed (beyond the probability distribution), the authors used a robust optimization method. An optimization framework for quickly finding routes for EVs was presented in [232] by M. Fontana;

the study was focused on total energy consumption during the minimum requested time, stating that the main routing problem in the current study was solved.

The influence of EVs on the market affects all infrastructures, and robust optimization may be also applied to the operation and planning of power systems, for example during charging of EVs. A. Souroudi et al. in [233] proposed a robust optimization based method for optimal charging/discharging of EVs considering the electricity price uncertainties.

It can be concluded that the optimization of an electric drive system is slightly dependent on the robustness of a single component; to achieve a robust drive system, a robust design optimization should be done at the system level. Different levels of electric drive designs are shown in Figure 9. As can be seen from the figure, a number of different sub-level optimizations must be considered together with the electrical machine; for example, the efficiency maps of the gearbox and inverter and the parameters of the controller.

The Digital Twin (DT) [234,235], as a physical replica of a system, presents a new approach in modeling and optimization, as well as in control, maintenance, diagnostics, and many other services [236]. For maximum performance, DT needs to have a specific algorithm to be characterized in virtual assets. While the life-cycle of the electrical drive as a product lasts several decades [237] , it is important for DT to be reliable during all of the exploitation period; therefore, special attention must be given to the applied software as well.

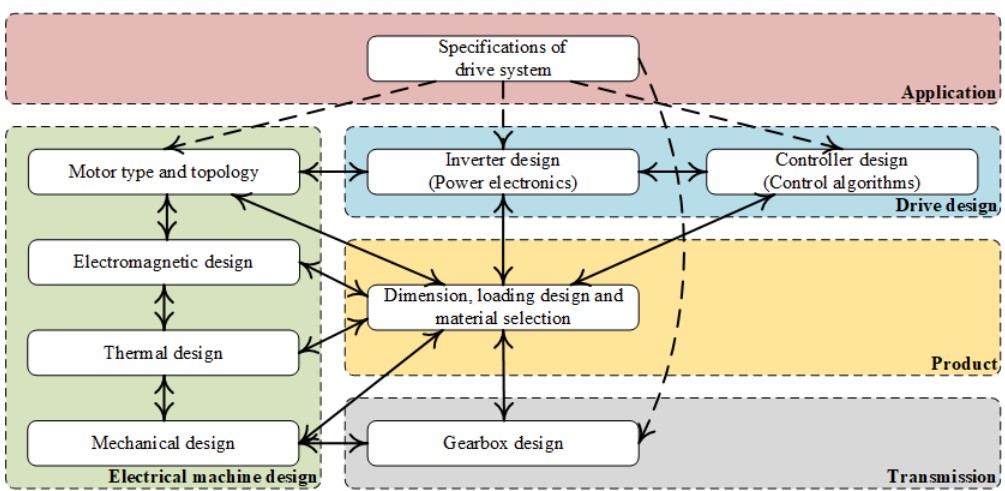

**Figure 9.** Complex framework of electrical drive system-level design optimization.

### *3.2. 3D Printed Electrical Machines*

One of the interesting research fields of electrical machines, where optimization plays a big role, is Additive Manufacturing (AM), also known as 3D printing (Figure 10). AM refers to a manufacturing process where a three-dimensional (3D) object is built in layers by depositing a material. This process allows making complex parts that would be difficult or even impossible to make with conventional manufacturing methods. Due to its advantages, AM technology is gaining more and more popularity in various applications. AM refers to a wide range of techniques as there are different methods to produce the 3D component. For example, the American Society for Testing and Standards (ASTM) has defined seven types of AM based on the manufacturing process [238]:

- vat photopolymerization;
- material jetting;
- binder jetting;
- material extrusion;
- powder bed fusion;
- sheet lamination;

- direct energy deposition.

From the electrical machine design point of view, AM offers several advantages. 3D printing offers the ability to produce complex three-dimensional shapes that can be used in every component of an electrical machine, as illustrated in Figure 10. It is possible to produce the frames of the machine using plastic or metal printers, which can result in lower weight or in better cooling capability. It is possible to design and manufacture a three-dimensional magnetic core for the stator and/or rotor, resulting in better usage of magnetic material or increasing the power density of the machine (Figure 11). It is possible to print electrically conductive materials, and this allows producing the windings of the machine. Through this, it is possible to increase the winding filling factor, increase the cooling capability, or reduce the end winding effect. It is also possible to produce special heat exchangers for the machine. Furthermore, it is possible to print out a full working electrical machine, resulting in rapid prototyping. This all sounds very good, but there is still much research work needed before all these advantages can be implemented in electrical machines.

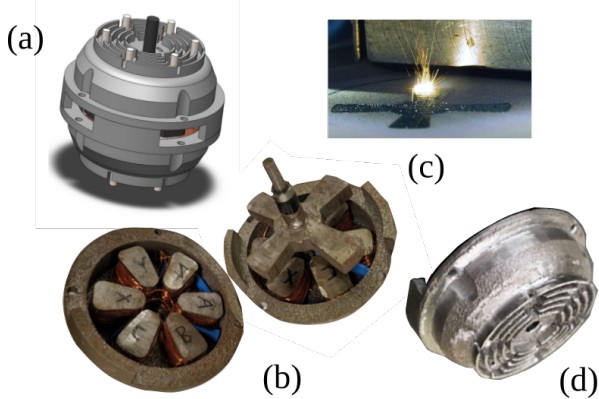

**Figure 10.** A model (**a**) and the geometry (**b**) of a 3D printed electrical machine. (**c**) shows one step from the design process, and (**d**) shows the printed, optimized heat exchanger.

The AM research community is ever-growing, developing new printing technologies and improving the existing ones to resolve its main drawbacks [239]: relatively slow manufacturing speed, internal defects from layer-by-layer fabrication, limited multi-material capabilities, and the necessary post-production of printed parts. From the material point of view, there is much work going on in the field of 3D printed materials. With magnetic materials, the main aim of most research work is to achieve material characteristics equal to or better than conventional materials [240–243]. From the magnetic point of view, there are two main things that need to be focused on: the material hysteresis loop and material conductivity, resulting in eddy current losses. One of the biggest challenges in using these materials in electrical machines is how to reduce the eddy current losses. One way to reduce these losses may be to groove the surfaces in which the eddy currents are formed, but this can greatly affect the average torque [244,245]. Another option could be to choose materials that have ferromagnetic properties, but lower conductivity. From the modeling point of view, the non-homogeneity of the printed materials and the unknown properties create new challenges for the designers [246]. However, there are new obstacles: 3D printed parts can suffer from internal defects; post-processing is often required; and key properties are often unknown about the materials from which it is printed. In addition, these materials often show anisotropy, both mechanically and electromagnetically. Therefore, if we want the resulting machine to be resistant to these phenomena, it is necessary to use robust optimization tools.

Furthermore, there is much work on conductive materials [247–249], where the main aim is to achieve the highest conductivity, and the main concerns are how to create the electric isolation around the wire. Not only the material characteristics are important, but also there are concerns with printing quality itself [250]. Most 3D printed objects need post-processing because of printing supports and

surface quality, but also to enhance the material properties. To produce a full electrical machine with 3D printing, there is a need for a multi-material metal printer. At the moment, the best quality is achieved with selective laser melting (SLM) , but this method allows printing only one material at a time. Other metal printing methods like binder jetting or direct energy deposition allow printing several materials at the same time, but with reduced material quality. There is an effort to reduce the post-processing (turning, milling, etc.); the surface quality can be worse than conventional manufacturing. Therefore, there is a higher concern about the manufacturing tolerances here.

The earliest examples of using a 3D printer in electrical machine manufacturing was printing the housing of the machine from Fused Deposition Modeling (FDM) and plastic [251]. This solution does not give much advantages to the machines, but it can show the possibilities of what can be done. In [252], a similar solution was used, but instead of using pure plastic, some parts were printed with iron powder. There are several possibilities to increase the cooling capability of the electrical machine through 3D printing. One possibility is to produce a special frame for the machine, where the cooling channels are printed inside the frame [253–255]. This allows improving the overall thermal performance of the machine. Instead of cooling the machine through the machine frame, it has been proposed to directly cool the machine windings. For this, printing a hollow winding or using direct winding heat exchangers has been proposed [256]. For example, in [257], a 3D printed direct winding heat exchanger was proposed that was installed in direct contact with the concentrated winding of a flux-switching permanent magnet machine. For this, they used the FDM printing method, and different polymers were used to print the heat exchanger. This allowed improving the power density of the machine. In [258], a similar heat exchanger was proposed, but printed with ceramic material.

The advantages of AM to produce thin layers and 3D shapes make it a good solution for producing lightweight machines. It allows reducing the weight by reducing the material used for machine supports [259]. For example, in [260], the authors presented a lightweight steel powder rotor where a lattice structure was used. Through this, they managed to reduce the machine weight by 25 %, as well as a 23 % inertia reduction. Furthermore, it is possible to produce the machine frame with a lattice structure, resulting in weight reduction. In [261], the authors reduced the volume of the magnetic core of a 3D printed magnetic clutch by 24% by increasing the magnetic force by more than 10% with the application of multiobjective robust design optimization methodologies (Figure 11). The possibility to produce complex 3D shapes allows more comprehensive optimization of the magnetic circuit, i.e., the stator and rotor cores of the machine, without the limitations of conventional subtractive and formative manufacturing methods, which can result in designs with significantly enhanced performance and notably lower material consumption and costs. For example in [262,263], the shape optimization of the reluctance machine rotor was presented, resulting in the minimization of the torque ripple and mass of the rotor. Furthermore, a 3D printed winding can be designed [264], resulting in weight reduction or a higher filling factor.

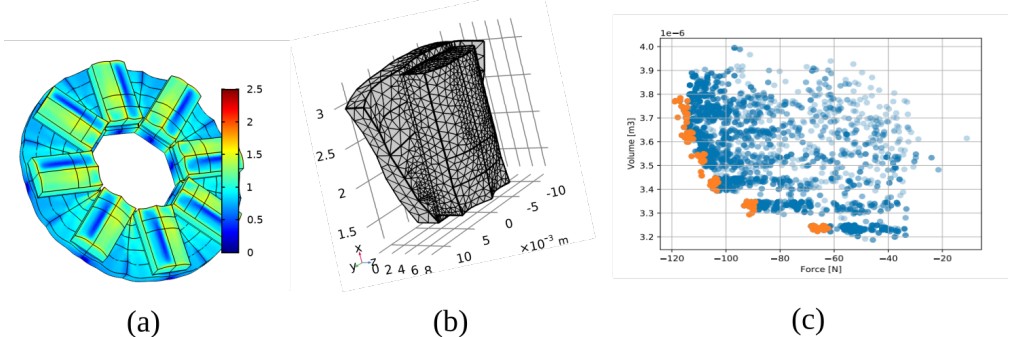

(a)                    (b)                    (c)

**Figure 11.** Example of design optimization and weight reduction of a 3D printed magnetic clutch with the Ārtap framework: (**a**) magnetic flux density, (**b**) is the mesh and (**c**) is the Pareto front.

### 3.3. Superconducting Material Based Electrical Machines

Superconducting materials have a set of particular electromagnetic properties. Under a certain critical temperature, their resistivity is negligible, and the magnetic flux density is expelled from the material. This perfect diamagnetism is called the Meissner effect, i.e., the exclusion of the magnetic field from the core of the superconducting material. High Temperature Superconducting (HTS) materials for power applications, such as electrical machines, are commercially available under the form of bulks and tapes. Bulks, mostly made of Yttrium-Barium-Copper-Oxide (YBCO) alloys, are obtained by a process known as Top-Seeded Melt Growth (TSMG), which consists of a method to fabricate single or multiple grain superconductors. The seeds, which act as nucleation centers, are placed on the top surface of YBCO compacted pellets and subjected to heat treatment processes, followed by melt texture growth heating cycles in order to meet the bulks. Unfortunately, the TSMG process is confined to small-scale and simple material geometries [265–267]. Besides scale limitations, bulks are susceptible to cracking due to irregular thermal expansion of their body, and therefore, long-term operation can be compromised [268].

In the case of tapes, most adequate for large-sized machines due to the long length of tape available and mechanical robustness, the most common compounds are Bismuth-Strontium-Calcium-Copper-Oxide (BSCCO), i.e., First-Generation (1G) tape, and YBCO, i.e., Second-Generation (2G) tape. The latter are best suited for working in higher magnetic fields than 1G tapes [269]. When compared to 1G tapes, 2G tapes take advantage of their improved properties, such as mechanical robustness, operation at a higher magnetic field, and critical current. These 2G coated conductors are manufactured by a continuous process using thin film deposition techniques in order to apply the superconducting material to buffered metal substrates. The manufacturing steps include electroplating, sputtering, electro-polishing, Ion Beam Assisted Deposition (IBAD), and Metal Organic Chemical Vapor Deposition (MOCVD). The IBAD process consists of the sputtering of a stack of buffer layers that develop a texture for the superconductor material's deposition. MOCVD is the deposition of the superconductor, based on Rare-Earth elements with Barium-Copper-Oxide (REBCO) materials. Further details concerning the manufacturing process can be found in [269–271].

On the other hand, Magnesium Diboride (MgB2) presents an alternative to YBCO and BSCCO [272]. Despite not being the natural choice since it requires lower cooling temperatures than YBCO or BSCCO, the MgB2 superconducting material benefits from lower prices. Commercial MgB2 is fabricated by a Powder-In-Tube (PIT) process. The manufacturing process was addressed in [273]. The wide availability of raw materials and decreasing fabrication costs lead to a trade-off between material price and cooling costs, and it is expected to compete with YBCO and BSCCO.

One of the main advantages of superconductors is the possibility of generating high magnetic flux densities with smaller amounts of material, when compared to conventional materials, such as copper coils or PMs [274]. Therefore, the design of electrical machines can benefit from these superconducting materials, allowing us to build smaller and lighter machines when compared to conventional ones. From DC machines (i.e., commutator and homopolar types) to AC machines (i.e., induction, homopolar, and synchronous), superconducting materials can be employed in every machine type [268]. By using a superconducting material for the excitation winding, field winding losses are practically negligible in DC conditions. In AC conditions, losses are reduced, but not negligible, mainly due to the AC loss phenomenon [275,276]. On the other hand, cryogenics can also present a design challenge since it can increase the size and costs of the device [277]. Therefore, besides the common constraints in machine design, including volume and material costs, cryogenics and the AC loss phenomenon also constitute design optimization challenges [167,278].

Synchronous machines are the most common topology in high-power applications (typically greater than 1000 hp ) [279]. For these machines, they can be coreless or ferromagnetic cores can be used. Generally, the design is supported by finite element modeling [280] and optimization algorithms [277]. One major concern to finite element modeling is the long computation time that is required.

The design optimization of superconducting machines includes the optimization of efficiency, the power factor, the length of superconducting windings, as well as machine and cryostat volumes. For the sake of cost savings, the length of superconducting coils and the machine volume should be reduced, whilst for the electrical characteristics, efficiency and the power factor should be as high as possible. A design example considering these criteria was presented in [281].

Modeling the Meissner effect and the related thermal properties of superconducting materials increases the complexity of the multiobjective optimization of superconducting material based electrical machines. A typical design problem considers, at least, the minimization of the superconducting winding material and the maximization of the power density, the solution of which generally requires a multiobjective algorithm [167]. Moreover, it is hard to formulate these design problems analytically without an FEM calculation. Furthermore, the superconducting materials are generally non-linear; their electrical and magnetic properties depend on their temperature, current density, and magnetic field. The applied FEM code has to handle this non-linearity as well. Another numerical problem is related to the thickness of the superconducting coating on the tapes, which are very thin (this can fall below the micrometer range). This can significantly increase the numerical complexity of the design problem. An hp-adaptive meshing strategy can significantly increase the accuracy of the calculation [22,282,283]. Due to the unique properties of the superconducting materials, there are novel possibilities. Design considerations should be made before they are applied to the electrical machine; it is not enough to follow the old design principles. For example, it is possible to reduce the short-circuit current in the conductors. Therefore, the short-circuit properties of large machines should be revised.

## 4. Conclusions

A wide variety of optimization tools are used for electrical machine design and analysis. The aim of these novel optimization methods is not to replace experienced designers; the goal is to provide practical tools and methods that can help in finding robust and high-performance solutions that are insensitive to the different kinds of manufacturing tolerances. Moreover, the novel optimization methods should handle the most recent technologies, such as additive manufacturing, superconductor based machines, and digital twins. This literature survey is different from the comparable studies in a number of important ways. Firstly, this paper highlights the importance of the "no free lunch" theorem of mathematical optimization and the specific benchmarks for the selection and comparison of nature-inspired solvers to an electrical machine optimization task. Moreover, the paper points out where the electrical machine design tasks are computationally intensive (multiobjective optimization tasks with many uncertainties) and which aspects are important during the selection and comparison of the available optimization methodologies. Although the meta-modeling techniques provide flexibility, in order to avoid information loss and false optima, it is essential to select the right methodology and accurate tools. The right choice can accelerate a wide variety of calculations Based on the modern electrical machine optimization frameworks, it is necessary to deal with hp-adaptive FEM solvers, evolutionary and genetic algorithms, parallelization, as well as meta-modeling techniques. All these techniques were discussed in this paper.

**Author Contributions:** Conceptualization, T.O.; methodology, T.O., A.R., A.K. and P.A.; formal analysis, T.O. and A.R.; investigation, T.O., D.P. and P.K.; resources, J.K. and D.P; data curation, J.K. and D.P.; writing—original draft preparation, T.O., A.R., A.K., P.A. and J.K.; writing—review and editing, T.O. and A.R.; visualization, D.P., T.O., J.K. and P.K.; supervision, D.P. and T.O.; project administration, T.O. and A.R.; funding acquisition, P.K., D.P. and A.R. All authors have read and agreed to the published version of the manuscript.

**Funding:** The research was supported by the Estonian Research Council under Grant PSG453 "Digital twin for propulsion drive of autonomous electric vehicle". This work was financially supported by Government of Russian Federation, Grant 08-08.

**Conflicts of Interest:** The authors declare no conflict of interest.

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
