# Peer review of "Robust Design Optimization and Emerging Technologies for Electrical Machines: Challenges and Open Problems"

_applsci, doi:10.3390/app10196653_

Round 1

Reviewer 1 Report

The paper deals with robust optimization in electrical machines manufacturing. It consists in a very large review of different technologies developed for this goal.

Unfortunatly there are, from my point of view , no sufficient quantitative and comparative developments .

Numerical tables should be added on  factors , values , uncertainties and tolerances for each technology or algorithm.

Algorithms ( l598) should be provided each time it is possible.

I think that the paragraphs on additive manifacturing  L450 and superconducting L537  seam to be out of scope in their present shapes and not in relationship with the other part of the paper.

From my point of view , the developments are too general and don't provide sufficient advances for the journal " Applied sciences "

Reviewer 2 Report

The paper deals with a very important problem of modern machine designs. Three main aspects are described: optimization techniques, new materials and additive manufacturing. The paper summarizes current achievements in these fields and includes a very rich literature. It could be very useful for both experienced and young engineers.

The paper is quite well written. However, some improvements needs to be introduce prior to publish the paper. I have marked them in the attached file. The major remarks are:

  • There are no references to figs. 5, 6, 10 and 11.
  • Page 6, last paragraph – something is missing.
  • English should be improved in many places. There are too many unclear and long sentences (I have marked them in the attached file).

Reviewer 3 Report

This paper comprehensively explores different methods for robust and optimum design of electric machines with a focus on manufacturing tolerances and state of the art manufacturing methods. It is quite an interesting read. The figures are particularly nice as they simply explain complicated concepts.

There are some minor typos and the paper overall needs proof reading, for example: “which is deals with simplified machine models” or “there is not exist a metaheuristic method”.

Two recommendations I have are:

  • Add tables listing several methods with a short description or important points for each. Using bullet points more freely would also help. The purpose of this recommendation is to make it easier for the reader to extract important information.

  • In several sections there are lists of different methods, for example robust design methodologies, or optimization algorithms. Although one cannot list them in the order of best to worst, it is possible to list them in the order of popularity, that is most and least frequently used. This can also be tabulated and would greatly add to the value of your research.

There are other reviews on robust design for electric machines. This manuscript is among good ones but needs some more work to really distinguish if from others. I see potential and believe the above two comments can help the authors to achieve that.
